# DASH: Deterministic Attention Scheduling for High-throughput Reproducible LLM Training

**Xinwei Qiang**[1,3]**, Hongmin Chen**[2]**, Shixuan Sun**[1]***, Jingwen Leng**[1]**, Xin Liu**[2]**, Minyi Guo**[1]
[1]School of Computer Science, Shanghai Jiao Tong University
[2]ByteDance Seed, [3]Zhiyuan College, Shanghai Jiao Tong University
[1]{qiangxinwei, sunshixuan, leng-jw, guo-my}@sjtu.edu.cn
[2]{chenhongmin.will, liuxin.ai}@bytedance.com

## Abstract

Determinism is indispensable for reproducibility in large language model (LLM) training, yet it often exacts a steep performance cost. In widely used attention implementations such as FlashAttention-3, the deterministic backward pass can incur up to a 37.9% throughput reduction relative to its non-deterministic counterpart, primarily because gradient accumulation operations must be serialized to guarantee numerical consistency. This performance loss stems from suboptimal scheduling of compute and gradient-reduction phases, leading to significant hardware underutilization.

To address this challenge, we formulate the backward pass of deterministic attention as a scheduling problem on a Directed Acyclic Graph (DAG) and derive schedules that minimize the critical path length. Building on this formulation, we present DASH(Deterministic Attention Scheduling for High-Throughput), which encapsulates two complementary scheduling strategies: (i) Descending Q-Tile Iteration, a reversed query-block traversal that shrinks pipeline stalls in causal attention, and (ii) Shift Scheduling, a theoretically optimal schedule within our DAG model that reduces pipeline stalls for both full and causal masks.

Our empirical evaluations on NVIDIA H800 GPUs demonstrate that DASH narrows the performance gap of deterministic attention. The proposed strategies improve the throughput of the attention backward pass by up to $1.28\times$ compared to the baseline, significantly advancing the efficiency of reproducible LLM training.

Our code is open-sourced at `https://github.com/SJTU-Liquid/deterministic-FA3`.

## 1 Introduction

The pursuit of consistent and verifiable outcomes is a cornerstone of rigorous scientific research and large-scale engineering. In the domain of large language model (LLM) training (Wu et al., 2024), where experiments span thousands of GPUs (Grattafiori et al., 2024; DeepSeek-AI et al., 2025) and incur enormous costs, this principle of reproducibility becomes indispensable. Reproducibility empowers practitioners to diagnose training instabilities, such as loss divergence, and to evaluate the impact of architectural modifications. Consequently, deterministic training, which guarantees bitwise identical results across runs, is increasingly adopted as a standard practice for industry.

The origin of the non-determinism in attention of LLM training can be traced back to a fundamental yet often overlooked characteristic of computer arithmetic: the non-associativity of floating-point (FP) operations (Villa et al., 2009). For instance, $(10^8 + 10^{-6}) - 10^8$ evaluates to 0.0 in single-precision, whereas $10^8 - 10^8 + 10^{-6}$ yields the correct $10^{-6}$. This sensitivity is magnified in the massively parallel environment of GPUs (Shanmugavelu et al., 2024).

---

*Shixuan Sun is the corresponding author.

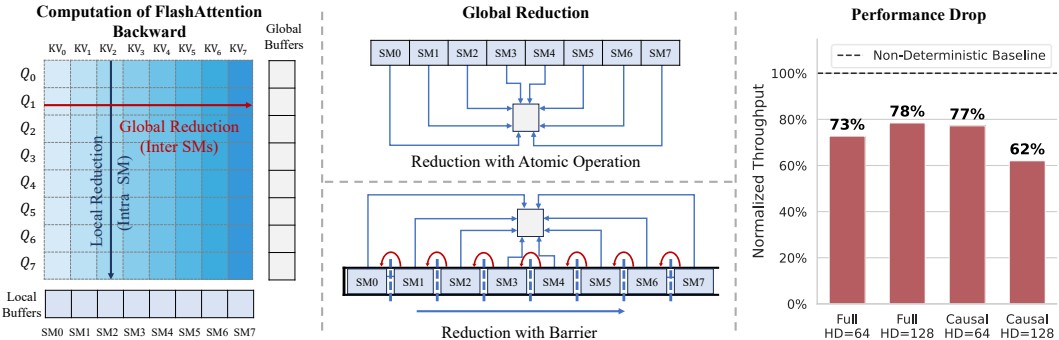

Figure 1: Overview of the deterministic FlashAttention. Left: Tiled computation structure of the backward pass, highlighting the local and global reductions. Middle: Comparison between the non-deterministic (atomic-based) and deterministic (ordered) global reduction. Right: Performance degradation under causal and full attention masks, HD stands for head dimension.

In high-performance attention (Vaswani et al., 2017) mechanisms like FlashAttention (Dao et al., 2022), the backward pass computation is parallelized across hundreds of GPU Streaming Multiprocessors (SMs) (NVIDIA, 2022) to maximize throughput. Each SM, running a Cooperative Thread Array (CTA), accumulates a partial contribution to gradient tensors (e.g., the gradient for the query matrix, d$Q$). The default high-speed approach allows these CTAs to concurrently update the final gradient in global memory via non-deterministic atomicAdd operations, as shown in Figure 1 middle. This creates a non-deterministic accumulation order: the final accumulated value depends on the uncontrolled completion order of the CTAs, leading to bit-wise variations between runs.

To enforce reproducibility, FlashAttention-3 (Shah et al., 2024) provides a deterministic mode. It enforces a fixed accumulation order by using synchronization barriers to force CTAs to perform their additions in a serialized order (e.g., ordered by CTA index). However, this guarantee of consistency imposes a significant performance penalty. As illustrated in Figure 1 right, enabling deterministic mode may lower throughput by up to 37.9%, leading to severe training costs when scaling LLMs across hundreds of thousands of GPUs.

This performance gap is not an inherent consequence of serialization itself. Instead, it stems from a direct conflict between the tile scheduling and a rigid, pre-determined accumulation order. As illustrated in the middle of Figure 1, the full mask scenario, commonly employed in multi-modal tasks, highlights a key inefficiency: the naive schedule creates a bottleneck by forcing reductions to start sequentially. An ideal schedule, however, would parallelize this process, allowing CTAs to begin reduction on different tiles concurrently. Crucially, this reveals that the computation schedule and the accumulation order are tightly coupled and cannot be optimized in isolation.

To address this, we introduce **D**eterministic **A**ttention **S**cheduling for **H**igh-throughput (DASH), a framework that formulates deterministic attention backward execution as an explicit scheduling optimization problem. We model the deterministic backward pass as a Directed Acyclic Graph (DAG), and formalize the objective as minimizing the DAG's critical path length. Based on this model, we design two complementary scheduling strategies. The first, *Descending Q-Tile Iteration*, is a heuristic that processes query tiles in reverse order to advance dependency resolution and shrink pipeline bubbles in causal attention. The second strategy, a theoretically optimal algorithm we term *Shift Scheduling* is provably optimal under our DAG model. It employs a phase-shifted assignment of computational tasks to GPU multiprocessors, creating a perfectly staggered execution pattern. This ensures that the workload is perfectly balanced and that the serialized reduction operations proceed without contention while approaching the model's theoretical utilization bound.

Our empirical evaluations on NVIDIA H800 GPUs show that DASH significantly narrows the performance gap relative to the FlashAttention-3 deterministic baseline. The two strategies deliver up to a $1.28\times$ speedup for the deterministic attention backward pass, significantly improving the efficiency of reproducible LLM training.

In summary, we made the following contributions in this paper:

- We identify the misalignment between tile execution and accumulation ordering as the principal source of performance degradation in deterministic attention.

- We provide the first DAG-based formalization of deterministic attention backward scheduling, enabling principled optimization of critical path length.

- We introduce two complementary scheduling strategies, Descending Q-Tile Iteration and Shift Scheduling, that achieve up to a $1.28\times$ speedup over the FlashAttention-3 deterministic baseline on H800 GPUs.

## 2 BACKGROUND

### 2.1 DETERMINISTIC FLASHATTENTION BACKWARD PASS

We first outline the core gradient computations in the FlashAttention backward pass: $dQ$, $dK$, and $dV$ (Figure 1, left). During backpropagation, the gradients $dK$ and $dV$ are accumulated across all queries for each key (or value) position, i.e., they are reduced along the $Q$ axis. In contrast, $dQ$ requires a reduction across all key–value (KV) positions for each query, i.e., along the KV axis. To expose parallelism, the implementation partitions the KV dimension across SMs, allowing $dK$ and $dV$ to be computed within each SM via a local reduction. However, this strategy distributes partial contributions to $dQ$ over multiple SMs, necessitating a global reduction to produce the final gradient. A conventional implementation performs this reduction using atomic additions (Figure 1, middle), which induces run-to-run variation because floating-point addition is non-associative. The resulting numerical nondeterminism undermines strict reproducibility in large-scale training. To guarantee determinism, one must enforce a prescribed accumulation order. FlashAttention-3 achieves this by performing a tile-wise sequential accumulation of $dQ$ along the KV dimension.

### 2.2 GPU ARCHITECTURE

On modern GPUs, the memory hierarchy comprises registers, shared memory, L2 cache, and global memory (NVIDIA, 2022), reflecting a fundamental capacity–latency trade-off: smaller and faster storage resides closer to the compute units. Shared memory is private to each SM, enabling low-latency intra-SM data reuse, whereas the L2 cache is globally shared, mediating inter-SM data exchange and coherence. In datacenter-class GPUs, the L2 cache may be physically segmented, with each segment preferentially serving a subset of SMs; remote-segment accesses typically incur higher latency than local ones. This hierarchical organization materially shapes the attainable performance and the efficiency of memory-bound GPU kernels.

### 2.3 DETERMINISM IN OTHER OPERATIONS OF THE TRANSFORMER

Other components, such as GEMMs, attention forward and normalization, also involve reduction operations; however, the computational cost of enforcing determinism in these cases is generally minimal during typical LLM training. GEMMs may exhibit nondeterministic behavior only when the reduction axis (i.e., the K-dimension) is partitioned across multiple blocks, as in split-K (NVIDIA Corporation, 2025) or stream-K (Osama et al., 2023) parallelization modes. In large-batch LLM training, parallelism along the M and N dimensions is typically sufficient to fully utilize the GPU, rendering split-K or stream-K modes unnecessary; therefore, disabling these modes generally results in only a minor reduction in throughput. Similarly, other operations involving reduction, such as attention forward passes and normalizations, typically perform reductions within a single block, thereby ensuring a deterministic reduction order. Purely elementwise operations, including activation functions and bias additions, are inherently deterministic.

## 3 DASH: SCHEDULING STRATEGIES FOR DETERMINISTIC ATTENTION

In this section, we introduce optimized scheduling strategies for deterministic attention. Without loss of generality, we assume that the number of KV tiles equals the number of SMs, denoted by $n$. When the actual number of KV tiles differs from the number of SMs, we conceptually refine or aggregate attention heads so that all SMs remain fully utilized under the same analytical framework.

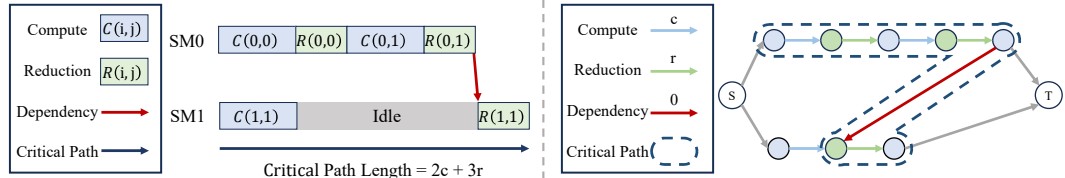

Figure 2: Visualization of the Deterministic Scheduling Problem. The Gantt chart (left) shows a naive execution schedule for a problem with two KV-tiles ($i$-index) and two Q-tiles ($j$-index). Each task consists of a compute phase $C(i, j)$ and a reduction phase $R(i, j)$. Local reductions enforce contiguous execution on a single SM (e.g., all tasks for $i = 0$ on SM0). A deterministic global reduction order introduces a cross-SM dependency (red arrow), forcing SM1 to idle and creating a pipeline bubble. The corresponding DAG (right) abstracts this schedule, where the critical path determines the end-to-end latency.

## 3.1 PROBLEM FORMULATION

We formalize the deterministic attention backward scheduling problem as an optimization over a directed acyclic graph (DAG), as shown in Figure 2. The DAG's structure is constrained jointly by the dataflow of FlashAttention and the architectural characteristics of the target GPU. Our model represents a simplified abstraction of actual GPU execution; its primary purpose is to offer insights into more effective scheduling decisions, rather than to accurately predict real execution times. As such, there remain significant differences between our theoretical model and the complexities of real-world GPU behavior.

**Graph Construction.** Each tile-processing task is modeled as a linear path of nodes connected by edges that encode two successive phases: (i) the tile's computation and (ii) the subsequent global reduction. These phase edges are weighted by their respective execution times, which are assumed to be constants. To encode legal accumulation orderings and data dependencies across tiles, we insert zero-weight dependency edges between nodes of different task paths. In this way, edge weights capture quantitative duration, while the topology captures qualitative ordering constraints. The scheduling objective is to minimize the critical-path length of the resulting DAG, thereby reducing end-to-end latency and improving overall execution efficiency.

**Optimization Constraint.** Data movement across different memory levels incurs substantial overhead, while registers provide the fastest storage in GPUs. To leverage fast register-resident accumulation of $\mathrm{d}K$ and $\mathrm{d}V$, all operations for a given KV tile must run contiguously on a single SM. Consequently, the edges associated with this tile form an unbroken chain, which imposes a key constraint on our optimization.

## 3.2 ANALYSIS OF FLASHATTENTION-3 DETERMINISTIC BACKWARD SCHEDULE

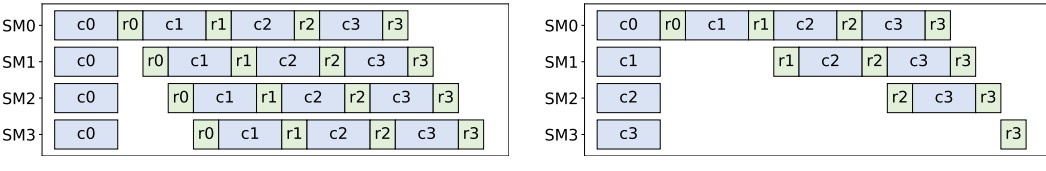

(a) Schedule for Full Mask        (b) Schedule for Causal Mask

Figure 3: Backward scheduling of FlashAttention-3 for both mask shapes. Each colored segment denotes one block's computation (cost $c$) followed by a reduction (cost $r$). Idle gaps correspond to pipeline bubbles. For clarity, since we assume the number of KV tiles equals the number of SMs, each SM processes exactly one KV tile; thus we omit the KV index in the visualization and show only the query index for each block.

Under a full attention mask, the FlashAttention-3 backward schedule achieves reasonable pipeline utilization (Figure 3a). Observable bubbles (SM idle periods) arise only during the startup phase of the first computation stage, before steady-state overlap is established. Let each stage incur a computation cost $c$ followed by a reduction cost $r$. After the initial fill, each attention head sustains $n$ sequential (computation + reduction) pairs, giving $T_{steady} = n \cdot (c + r)$ where $n$ is the number of

SMs. The startup overhead contributes an additional $(n-1) \cdot r$ due to staggered completion of the first sequence of reductions. Hence, for $m$ heads, $T_{full} = mT_{steady} + T_{startup} = m \cdot n \cdot (c+r) + (n-1) \cdot r$, up to negligible control and synchronization overhead.

In contrast, when a causal mask is applied, the data dependencies inherent in the schedule lead to significant inefficiencies. As shown in Figure 3b, this schedule introduces a substantial bubble within the execution of **each** attention head, preventing effective pipelining. The critical path for a single head becomes $T_{head\_causal} = n \cdot (c+r) + (n-1) \cdot r$. Since this inefficient pattern repeats for every head, the total execution time for $m$ heads is approximately $T_{causal} = m \cdot T_{head\_causal} + T_{startup} \approx m \cdot n \cdot (c+r) + (n-1) \cdot r$.

### 3.3 Descending Q-Tile Iteration: A Robust Heuristic for Causal Masks

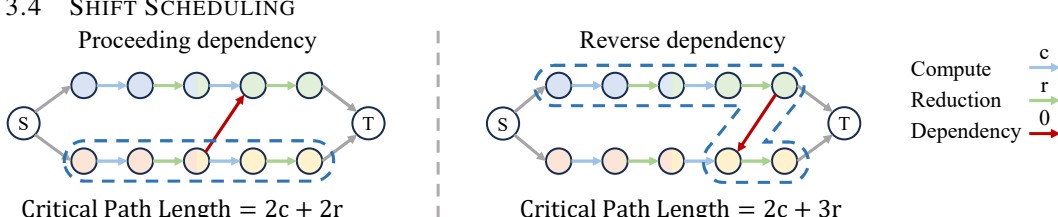

Figure 4: Descending (reverse-order) query tile schedule for the causal mask. Reversing the $Q$-block traversal accelerates dependency resolution. Colors distinguish attention heads in the pipeline.

To mitigate the pipeline bubbles caused by causal masking, we propose a simple yet effective modification: reversing the processing order of the query ($Q$) blocks. As illustrated in Figure 4, this reversed schedule allows most SMs to begin their computation earlier by resolving dependencies more quickly.

The crucial advantage of this approach is its impact on pipeline efficiency for subsequent attention heads. By reversing the order, the short tasks are completed first, freeing up their SMs much earlier. Consequently, the second head can immediately begin to utilize these available resources, creating a tightly coupled pipeline that almost eliminates the idle gaps between heads. This sustained high utilization across an even number of $m$ heads yields a total execution time of:
$$T_{reversed} \approx \frac{m \cdot (n+1)(c+r)}{2} + (n-1) \cdot r.$$

### 3.4 Shift Scheduling

Figure 5: Illustrative example for Lemma 1. Left: Added dependency (zero-weight) edges preserve non-decreasing depth order and do not lengthen the critical path. Right: A backward (depth-decreasing) dependency edge violates the lemma's condition and increases the critical path.

Although the Descending Q-Tile Iteration significantly improves performance, it is natural to ask whether a theoretically optimal schedule exists. To address this, we examine the impact of introducing reduction-induced inter-SM dependencies on the computation DAG's critical path.

Disregarding (for the moment) the accumulation edges required for d$Q$ updates, the graph decomposes into $n$ independent chains whose total time is minimized when their cumulative workloads are perfectly balanced. In this idealized scenario, all chains are also isomorphic, as they share an identical task structure and number of tasks. The core challenge is thus to insert the necessary zero-weight dependency edges without lengthening the original critical path. The lemma below characterizes precisely when this is possible; its proof is deferred to Appendix B for brevity.

**lemma 1.** *Let $G_0 = (V, E_0)$ be a DAG consisting of a single source node $s$, a single sink node $t$, and $n \geq 1$ parallel, isomorphic chains connecting $s$ to $t$. All edge weights in $E_0$ are strictly positive. Let the depth of a node $v$, denoted $depth(v)$, be the number of edges on the unique path from $s$ to $v$ within its chain in $G_0$. Let a sequence of graphs $G_1, \ldots, G_k$ be generated such that $G_i = (V, E_{i-1} \cup \{e_i\})$,*

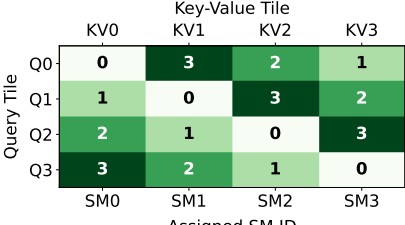

Figure 6: Optimal full-mask schedule via cyclic shifting. Left: Cyclic visiting order of $Q$ tiles per SM; distinct timestamps (i.e., the value in each box) on each row induce a natural, conflict-free reduction sequence for every $\mathrm{d}Q$ block. Right: Simulated timeline showing fully balanced utilization without additional bubbles.

*where each $e_i = (u_i, v_i)$ is a zero-weight edge. We add the explicit condition that every new graph $G_i$ in the sequence must remain a DAG.*

*Under this condition, the critical path length of $G_k$ is equal to that of $G_0$ if and only if for every added edge $e_i = (u_i, v_i)$ for $i \in \{1, \ldots, k\}$, the condition $depth(u_i) \leq depth(v_i)$ holds.*

As illustrated in Figure 5, Lemma 1 dictates that to preserve the original critical path length, any added dependency edge $(u, v)$ must satisfy the condition $depth(u) \leq depth(v)$. This formal constraint translates to a critical physical limitation: for any given query tile $Q_j$, the tasks involving it cannot be executed in parallel.

A schedule that assigns two tiles contributing to the same $\mathrm{d}Q_j$—say $(KV_i, Q_j)$ and $(KV_k, Q_j)$—to execute concurrently on different SMs would create a resource conflict during their reduction phases. Resolving this conflict requires serializing the reductions, for instance, forcing the reduction for $(KV_k, Q_j)$ to wait for the one from $(KV_i, Q_j)$ to complete, or vice versa. Because the conflicting reduction tasks would otherwise start at the same depth in the DAG, this forced serialization introduces a dependency edge $(u, v)$ where $depth(u) > depth(v)$—from the completion of the first reduction to the start of the second. This directly violates the lemma's condition and sub-optimally extends the critical path.

Our objective is thus twofold: first, to balance the workload across SMs, and second, to devise a conflict-free reduction order that adheres to the lemma's constraint.

**Optimal Schedule for Full Masks**    Under a full mask, per-KV-tile workloads are uniform, allowing for immediate balancing. To satisfy the second objective, we employ a *Shift Scheduling*, as illustrated in Figure 6. In this schedule, $SM_i$ processes KV blocks in the order $(i, i + 1, \ldots, n - 1, 0, \ldots, i - 1)$. This cyclical assignment inherently creates a conflict-free, sequential ordering for the reductions on any given $\mathrm{d}Q$ block, directly satisfying the lemma's condition. As both workload balancing and conflict-free reduction are achieved, this schedule is theoretically optimal.

**Symmetric Shift Scheduling for Causal Masks**    Causal masking induces a strongly imbalanced workload: early KV blocks participate in the full set of query interactions, whereas later blocks contribute progressively fewer operations, yielding computation workloads that decrease linearly across the sequence.

We address this by *Symmetric Shift Scheduling*. Its core is a symmetric pairing principle: SMs jointly handle KV blocks $i$ and $n - 1 - i$, pairing the longest with the shortest, the second-longest with the second-shortest, and so forth. This pairing equalizes task chain lengths per SM, restoring near-perfect balance.

We operationalize symmetric pairing via a two-phase schedule. In Phase 1, a cyclic shift is applied to the dense lower-left rectangle, efficiently filling the pipeline. Phase 2 addresses the residual triangles using a purely analytical model of workload folding, where tasks from the lower-right are logically mapped to the upper-left's masked slots to form a conceptual square without any data movement. The operational sequence—a top-down traversal of the left triangle and a bottom-up traversal of the right—is algebraically equivalent to a diagonal-initialized shift schedule on this conceptual square. This equivalence is key: it preserves workload balance, ensures contiguous computation for each

KV block, enforces depth-monotone accumulation to satisfy Lemma 1, and ultimately eliminates all pipeline bubbles.

**Summary of Optimal Performance**   In summary, the proposed scheduling strategies achieve theoretical optimality for both scenarios. By perfectly balancing workloads and eliminating pipeline bubbles, the total execution time for $m$ heads is: Full Mask: $T_{full\_opt} = m \cdot n \cdot (c + r)$; Causal Mask: $T_{causal\_opt} = \frac{m \cdot (n+1) \cdot (c+r)}{2}$

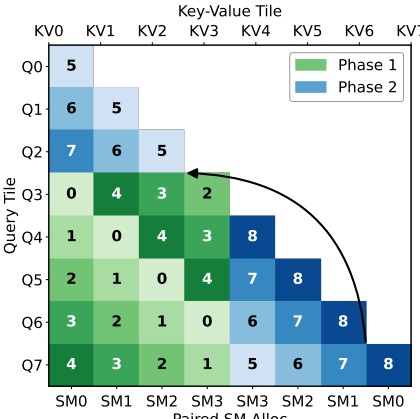

Figure 7: Optimal causal-mask schedule using symmetric shift and two-phase workload folding. Phase 1 processes the dense lower-left rectangle; Phase 2 folds the remaining triangles into a logical square and traverses it starting from the main diagonal, first covering the upper-left portion before the lower-right, ensuring each KV block is executed contiguously.

## 4   EXPERIMENTS

In this section, we empirically evaluate the performance of our proposed scheduling strategies under full and causal masks. We measure throughput under various sequence lengths and analyze how architectural factors interact with different scheduling choices.

### 4.1   EXPERIMENTAL SETUP

**Hardware and Software.**   All experiments are conducted on a server equipped with NVIDIA H800 GPUs, CUDA version 12.6 and Triton (Tillet et al., 2019) version 3.4. All kernels are implemented by extending the FlashAttention-3 implementation.

**Baseline and Proposed Methods.**   We compare our methods against the deterministic backward pass of FlashAttention-3, which serves as our primary baseline. We also benchmark against the Triton tutorial's implementation for causal attention, as its public version lacks a full-mask counterpart. We omit FlashAttention-2 because prior published benchmarks (Shah et al., 2024) on Hopper-class GPUs show it is consistently outperformed by FlashAttention-3, and thus it no longer constitutes a competitive baseline. The methods under our evaluation are:

- **Descending Q-Tile Iteration** (for both masks)
- **Shift Scheduling** (for full masks)
- **Symmetric Shift Scheduling** (for causal masks)

**Benchmark Settings**   Following the methodology of the FlashAttention-3 study, we evaluate performance by fixing the total number of tokens at 16,384 while varying the sequence length from 512 to 16,384. Similarly, we fix the hidden dimension to be 2,048, and test different head dimensions in 64 and 128. All the results are tested using BF16 precision random inputs.

### 4.2   PERFORMANCE ON FULL ATTENTION MASKS

Figure 8 presents the throughput comparison for the full attention mask scenario. Our Shift Scheduling consistently outperforms the FlashAttention-3 baseline across most sequence lengths, demonstrating the effectiveness of our theoretically optimal approach. However, a notable exception occurs at the maximum sequence length of 16,384, where its performance slightly degrades relative to the original FlashAttention-3 baseline.

This phenomenon highlights a divergence between our theoretical model and practical hardware execution. Our model assumes zero-cost dependency edges, but in reality, inter-SM communication for synchronizing reduction operations is mediated by the L2 cache. This incurs significant latency, ranging from approximately 200 cycles for accesses to the local L2 cache segment to over 500 cycles

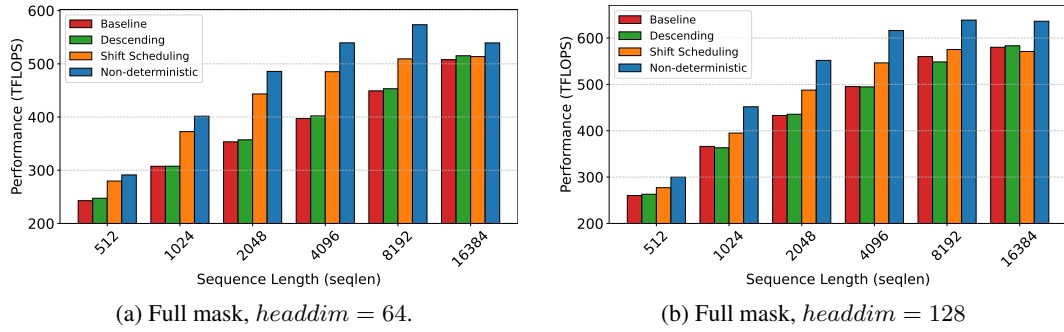

Figure 8: Backward-pass throughput under full attention masks.

for remote segment accesses on H800-class GPUs (Luo et al., 2025). This latency differential is a direct consequence of the distributed L2 cache architecture described in Section 2.

At a sequence length of 16,384 and a KV block size of 128, the computation for a single head is distributed across 128 blocks, often mapped to 128 SMs. This high degree of parallelism necessitates frequent cross-SM communication to signal task completion. Given the large number of participating SMs, a substantial portion of these synchronization signals must traverse the higher-latency links to a remote L2 cache segment. The Shift Scheduling, with its more intricate dependency graph compared to the simpler, linear dependency of the baseline, becomes more sensitive to this communication overhead at extreme parallelism. This increased synchronization cost, dominated by remote L2 accesses, ultimately outweighs the computational benefits of the schedule in this specific high-parallelism, long-sequence scenario, leading to the observed performance degradation.

## 4.3 PERFORMANCE ON CAUSAL ATTENTION MASKS

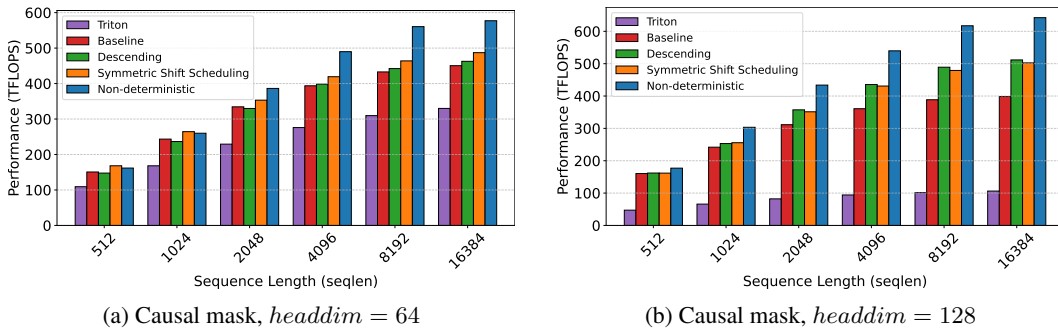

Figure 9: Backward-pass throughput under causal attention masks.

The performance evaluation for causal attention masks, presented in Figure 9, confirms the efficacy of our proposed methods. Both the Descending Q-Tile Iteration and our theoretically optimal Symmetric Shift Scheduling demonstrate a throughput improvement over the FlashAttention-3 baseline across all tested configurations.

An interesting trade-off emerges when comparing our two proposed methods at different head dimensions (headdim). At $headdim = 64$, the Symmetric Shift Scheduling achieves the highest performance, validating the benefits of its superior workload balancing. However, the descending schedule does not perform very well in this case. This is because in the FlashAttention-3 causal backward kernel, the L2-aware LPT scheduler interleaves multiple heads across SMs. When headdim = 64 and the sequence length is short, each head's L2 footprint remains small, allowing many heads to reside in cache with only 1–2 tiles in flight per head. Consequently, the causal stalls targeted by Descending Q-Tile Iteration are largely masked by cross-head interleaving, resulting in only marginal net performance gains.

However, at $headdim = 128$, Symmetric Shift Scheduling's performance is surpassed by the simpler Descending Q-Tile Iteration. This performance inversion is attributable to a critical interaction between algorithmic complexity and GPU resource limitations, specifically register pressure. The Symmetric Shift Scheduling, while algorithmically optimal, requires a more complex implementation to manage the state of the folded task space. This complexity translates to higher register usage per thread to maintain additional loop counters and intermediate states.

When $headdim = 128$, the base register requirement for storing accumulators and other intermediate values is already substantial. The additional overhead (around 10 registers) from our optimal schedule can push the total register count per thread beyond the hardware's physical limit, as shown by Nsight Compute (NVIDIA Corporation, 2024). This forces the compiler to generate code that spills registers, offloading their contents to the much slower local memory. The high latency incurred by these spill-induced memory operations introduces significant execution stalls, which negate the algorithmic benefits of the more balanced workload and lead to degraded performance. In contrast, the simpler Descending Q-Tile Iteration operates below this critical register pressure threshold, thereby avoiding spilling and achieving better effective performance in this high-resource-demand scenario. Therefore, the two schedules for causal masks are complementary: Symmetric Shift is theoretically optimal under our DAG model, while Descending is the practically preferred choice for large head dimensions on current GPUs.

In the future, Symmetric Shift's theoretical advantages are expected to be fully realized on newer architectures with greater on-chip resources (such as Blackwell GPUs with TMEM, or devices equipped with larger register files), or under kernel designs that are less constrained by register allocation than the present FlashAttention-3 implementation.

## 4.4 END-TO-END PERFORMANCE

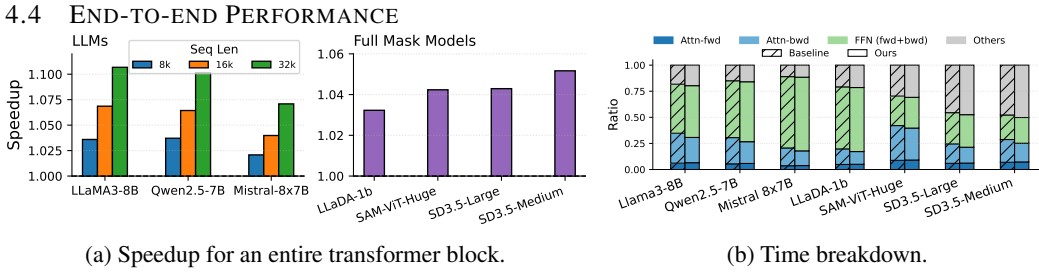

(a) Speedup for an entire transformer block.  (b) Time breakdown.

Figure 10: End-to-end performance of a transformer block.

To assess the performance gains delivered by DASH during training, we measured the runtime required to process an entire transformer block, accounting for both forward and backward passes.

We evaluated DASH across a range of widely adopted models. For causal mask scenarios, we selected famous LLMs: LLaMA3-8b (Grattafiori et al., 2024), Qwen2.5-7b (Qwen et al., 2025), and Mistral-8×7b (Jiang et al., 2024). For full mask scenarios, we included the vision model SAM-huge (Kirillov et al., 2023), the diffusion models StableDiffusion3.5 (medium and large) (Stability AI, 2024), and the diffusion-based language model LLaDA-1b (Nie et al., 2025).

For LLMs, we employ a batch size of 1 with sequence lengths of 8k, 16k, and 32k. In the case of full mask models, a batch size of 16 is used, with the training sequence length fixed at 4k in accordance with standard architectural configurations. The relative speedup achieved by our approach compared to the baseline is illustrated in Figure 10a. For causal models, we observe end-to-end performance improvements ranging from 2% to 10%. Full mask models also exhibit a speedup of approximately 4%. In summary we achieved an average speedup of around 5%, which aligns with our internal training experience on thousands of GPUs. Additionally, Figure 10b provides a detailed breakdown of computation time across different kernel operations, with causal models evaluated at a sequence length of 16k.

## 4.5 IMPACT OF DETERMINISM ON NUMERICAL STABILITY

Our analysis of backward passes indicates that non-deterministic kernels cause run-to-run gradient deviations of $O(10^{-4})$, while deterministic ones guarantee bitwise identical outcomes (Table 1). Although small, this variability can accumulate, so determinism is key to achieving reproducibility.

Table 1: Max gradient deviation averaged over 10 identical backward passes; $M_r = \max |g_r - g_{\text{ref}}|$.

| Masking Scheme | Non-deterministic | Deterministic |
|---|---|---|
| **Full** | $2.4 \times 10^{-4}$ | 0 |
| **Causal** | $4.9 \times 10^{-4}$ | 0 |

## 5 RELATED WORKS

**FlashAttention and Kernel-Level I/O Optimization**   Early optimization of attention focused on mitigating the I/O bottleneck imposed by the quadratic attention matrix. FlashAttention (Dao et al., 2022) introduced an I/O-aware tiled and fused kernel that avoids materializing the full attention matrix in HBM. FlashAttention-2 and 3 (Dao, 2023; Shah et al., 2024) further improved utilization via refined work partitioning and leveraged specialized hardware for asynchronous data movement.

**Low-Precision Attention**   Low-precision methods further reduce bandwidth and memory cost. The SageAttention series (Zhang et al., 2025b;a;c) systematically explores progressively lower formats while maintaining accuracy.

**Inference-Oriented Attention Kernels**   Inference-specialized kernels include FlashDecoding and FlashDecoding++(Dao et al., 2023; Hong et al., 2024) for autoregressive decoding, PodAttention(Kamath et al., 2025) for mixed prefilling/decoding, and DeFT (Yao et al., 2025) and Fast-Tree (Pan et al., 2025) for tree-structured generation.

**Distributed Cyclic Scheduling**   Our shift-based scheduling is inspired by cyclic (ring-style) phase-shift patterns long used in distributed systems.   Distributed attention algorithms—RingAttention (Liu et al., 2023), StripedAttention (Brandon et al., 2023), and Loong-Train (Gu et al., 2024)—adopt related cyclic schemes to overlap communication and computation across devices, whereas we apply a shift strategy intra-GPU to co-optimize deterministic accumulation and work balance.

**Deterministic Implementations**   Existing deterministic implementations either split $dK, dV$ and $dQ$ calculation into different passes (e.g., Triton tutorials (Tillet et al., 2019))—forcing a second K/V read—or materialize per-tile dQ partials for later consolidation (FlashAttention-2), adding memory footprint and an extra reduction kernel. These designs trade bandwidth or memory rather than co-optimizing execution and accumulation order, which is the focus of our approach.

**Determinism in Inference**   Determinism for inference has also been examined: He & Lab (2025) attribute non-reproducibility to lack of "batch invariance," where outputs depend on batch size, and design batch-invariant kernels. Their goal differs from ours: we target training time run-to-run determinism, where batch configurations are fixed to ensure reproducibility.

## 6 CONCLUSION

In this work, we addressed the significant performance penalty associated with the deterministic backward pass in modern attention mechanisms. By formulating the computation as a scheduling problem on a DAG, we introduced DASH, a framework featuring two distinct and complementary strategies. The first, Descending Q-Tile Iteration, provides a simple yet remarkably effective heuristic that accelerates causal attention. The second, derived from our conflict-free scheduling lemma, represents a theoretically optimal solution.

Our empirical evaluation not only demonstrates that DASH significantly narrows the performance gap, improving throughput by up to $1.28\times$ over the baseline, but more importantly, it reveals a crucial insight: theoretical optimality does not always translate to practical superiority. We identified hardware realities, such as register pressure and inter-SM communication latency, as critical factors that can override the benefits of a more complex, algorithmically perfect schedule. By providing a suite of solutions catering to different scenarios, DASH enables practitioners to achieve high throughput attention in reproducible LLM training.

## ACKNOWLEDGMENTS

This work is partially supported by Fundamental and Interdisciplinary Disciplines Breakthrough Plan of the Ministry of Education of China (JYB2025XDXM113).

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

## A  THE USE OF LARGE LANGUAGE MODELS

During the preparation of this manuscript, the authors employed a large language model (LLM) for two primary purposes. First, the LLM was used as a tool to improve the grammar, spelling, and overall clarity of the text. Second, it was used to assist in the initial stages of the literature search. The role of the LLM was strictly that of an assistant. All language suggestions were reviewed and edited by the authors to ensure they accurately reflected the intended scientific meaning. Furthermore, any literature identified with the assistance of the LLM was independently retrieved, reviewed, and vetted for relevance and accuracy by the authors. All intellectual contributions, including the conception of the research, methodology, and final conclusions, are the exclusive work of the human authors, who take full responsibility for the final content of this paper.

## B  PROOF OF LEMMA 1

*Proof.* Let $LP_i(x)$ denote the length of the longest path from the source node $s$ to node $x$ in graph $G_i$. The critical path length of $G_i$ is $CP(G_i) = LP_i(t)$.

Due to the isomorphic structure of the chains in $G_0$, all nodes at the same depth $j$ have the same longest path length from $s$. Let's denote this common length as $L_j = LP_0(v)$ for any node $v$ with $depth(v) = j$. Since all original edge weights in $E_0$ are strictly positive, it follows that for any two depths $j_1$ and $j_2$, if $j_1 < j_2$, then $L_{j_1} < L_{j_2}$. This implies $j_1 \leq j_2 \iff L_{j_1} \leq L_{j_2}$.

The proof proceeds by induction on the number of added edges, $k$.

**Base Case (k=1):** We prove the statement for the addition of a single edge $e_1 = (u, v)$ to $G_0$ to form $G_1$.

**Sufficient Condition ( $\implies$ ):** Assume $depth(u) \leq depth(v)$. By the lemma's premise, we are given that adding $e_1$ results in $G_1$ being a DAG. We must show that $CP(G_1) = CP(G_0)$.

The longest path to any node $x$ in $G_1$ is given by the recurrence $LP_1(x) = \max_{(w,x) \in E_1}\{LP_1(w) + weight(w, x)\}$. For node $v$, this becomes:

$$LP_1(v) = \max(LP_0(v), LP_0(u) + 0)$$

By definition, $LP_0(v) = L_{depth(v)}$ and $LP_0(u) = L_{depth(u)}$. The condition $depth(u) \leq depth(v)$ implies $L_{depth(u)} \leq L_{depth(v)}$. Thus, $LP_1(v) = \max(L_{depth(v)}, L_{depth(u)}) = L_{depth(v)} = LP_0(v)$. Since the longest path to $v$ is unchanged, and this is the only modification, the longest paths to all successors of $v$ also remain unchanged. Therefore, $LP_1(x) = LP_0(x)$ for all $x \in V$, which implies $CP(G_1) = CP(G_0)$.

**Necessary Condition ($\impliedby$):** Assume $CP(G_1) = CP(G_0)$ and (as per the lemma's premise) $G_1$ is a DAG. We prove the contrapositive: if $depth(u) > depth(v)$, then $CP(G_1) > CP(G_0)$.

Since $G_1$ is a DAG, adding the edge $(u, v)$ did not create a cycle. The longest path to $v$ becomes:

$$LP_1(v) = \max(LP_0(v), LP_0(u) + 0) = \max(L_{depth(v)}, L_{depth(u)})$$

Since we assume $depth(u) > depth(v)$ and all original edge weights are strictly positive, we have $L_{depth(u)} > L_{depth(v)}$. This leads to $LP_1(v) = L_{depth(u)} > L_{depth(v)} = LP_0(v)$. The longest path to $v$ has strictly increased. This increase propagates to all successors of $v$, including the sink $t$. Therefore, $LP_1(t) > LP_0(t)$, which means $CP(G_1) > CP(G_0)$. This contradicts our assumption. Thus, the condition $depth(u) \leq depth(v)$ is necessary.

**Inductive Hypothesis (IH):** Assume for some $k \geq 1$, the lemma holds. That is, given that $G_k$ is a DAG, $CP(G_k) = CP(G_0)$ if and only if the condition $depth(u_i) \leq depth(v_i)$ held for all $i \in \{1, \ldots, k\}$. We make the stronger hypothesis that if the condition held, then $LP_k(x) = LP_0(x)$ for all nodes $x \in V$.

**Inductive Step:** We prove the lemma for the addition of the $(k + 1)$-th edge, $e_{k+1} = (u, v)$, to $G_k$ to form $G_{k+1}$.

**Sufficient Condition ( $\implies$ ):** Assume $depth(u) \leq depth(v)$. By the lemma's premise, we are given that $G_{k+1}$ is a DAG. We must show $CP(G_{k+1}) = CP(G_k)$.

The longest path to $v$ in $G_{k+1}$ is $LP_{k+1}(v) = \max(LP_k(v), LP_k(u)+0)$. By the IH, since the conditions held for the first $k$ edges, we have $LP_k(v) = LP_0(v) = L_{depth(v)}$ and $LP_k(u) = LP_0(u) = L_{depth(u)}$. The calculation is identical to the base case: $LP_{k+1}(v) = \max(L_{depth(v)}, L_{depth(u)}) = L_{depth(v)} = LP_k(v)$. The longest path to $v$ is unchanged, and by propagation, $LP_{k+1}(x) = LP_k(x)$ for all $x \in V$. This maintains our strong hypothesis and proves $CP(G_{k+1}) = CP(G_k) = CP(G_0)$.

**Necessary Condition ($\impliedby$):** Assume $CP(G_{k+1}) = CP(G_k)$ and (as per the lemma's premise) $G_{k+1}$ is a DAG. We prove the contrapositive: if $depth(u) > depth(v)$, then $CP(G_{k+1}) > CP(G_k)$.

Since $G_{k+1}$ is a DAG, adding $(u, v)$ did not create a cycle. We compute $LP_{k+1}(v)$:

$$LP_{k+1}(v) = \max(LP_k(v), LP_k(u) + 0)$$

Using the IH ($CP(G_k) = CP(G_0)$ implies the conditions held for the first $k$ edges, so $LP_k(x) = LP_0(x)$ for all $x$):

$$LP_{k+1}(v) = \max(LP_0(v), LP_0(u)) = \max(L_{depth(v)}, L_{depth(u)})$$

Since we assume $depth(u) > depth(v)$, we have $L_{depth(u)} > L_{depth(v)}$. This leads to $LP_{k+1}(v) = L_{depth(u)} > L_{depth(v)} = LP_0(v) = LP_k(v)$. The longest path to $v$ strictly increases. This increase propagates to the sink node $t$, so $CP(G_{k+1}) > CP(G_k)$. This contradicts our assumption. Therefore, the condition is necessary.

By the principle of induction, the lemma holds for any $k \geq 1$. $\square$

## C  EXACT ALGORITHM AND MODIFICATIONS

We present the exact algorithm in Algorithm 1 in this section. The following pseudocode is adapted from the original FlashAttention-3 paper (Shah et al., 2024), with all modifications introduced by DASH explicitly marked.

---

**Algorithm 1** DASH algorithm

---

**Require:** Matrices $\mathbf{Q}, \mathbf{K}, \mathbf{V}, \mathbf{O}, \mathbf{dO} \in \mathbb{R}^{N \times d}$ in HBM, logsumexp vector $L \in \mathbb{R}^N$ in HBM, block sizes $B_c$, $B_r$.

1: In a preprocessing kernel, compute $D = \text{rowsum}(\mathbf{dO} \circ \mathbf{O}) \in \mathbb{R}^d$ (pointwise multiply), write $D$ to HBM and divide it into $T_r$ blocks $D_1, \ldots, D_{T_r}$ of size $B_r$ each.

2: Divide $\mathbf{Q}$ into $T_r = \left\lceil \frac{N}{B_r} \right\rceil$ blocks $\mathbf{Q}_1, \ldots, \mathbf{Q}_{T_r}$ of size $B_r \times d$ each, and divide $\mathbf{K}, \mathbf{V}$ in to $T_c = \left\lceil \frac{N}{B_c} \right\rceil$ blocks $\mathbf{K}_1, \ldots, \mathbf{K}_{T_c}$ and $\mathbf{V}_1, \ldots, \mathbf{V}_{T_c}$, of size $B_c \times d$ each.

3: Divide $\mathbf{dO}$ into $T_r$ blocks $\mathbf{dO}_i, \ldots, \mathbf{dO}_{T_r}$ of size $B_r \times d$ each, and divide $L$ into $T_r$ blocks $L_i, \ldots, L_{T_r}$ of size $B_r$ each.

4: Initialize pipeline object to manage barrier synchronization with $s$-stage circular SMEM buffer.

5: **if** in producer warpgroup **then**
6:     Deallocate predetermined number of registers.
7:     Issue load $\mathbf{K}_j$ and $\mathbf{V}_j$ from HBM to shared memory.
8:     Upon completion, commit to notify consumer of the load of $\mathbf{K}_j$ and $\mathbf{V}_j$.
9:     **for** i in assigned Q-tile schedule **do** **[DASH]**
10:         Wait for the $(i \% s)$th stage of the buffer to be consumed.
11:         Issue loads of $\mathbf{Q}_i, \mathbf{dO}_i$ from HBM to shared memory at the $(i \% s)$th stage of the buffer.
12:         Upon completion, commit to notify consumers of the loads of $\mathbf{Q}_i, \mathbf{dO}_i$.
13:     **end for**
14: **else if** in consumer warpgroups **then**
15:     Reallocate predetermined number of registers as function of number of consumer warps.
16:     On-chip, Initialize $\mathbf{dK}_j = (0)_{B_c \times d}, \mathbf{dV}_j = (0)_{B_c \times d}$ .
17:     Wait for $\mathbf{K}_j$ and $\mathbf{V}_j$ to be loaded in shared memory.
18:     **for** i in assigned Q-tile schedule **do** **[DASH]**
19:         Wait for $\mathbf{Q}_i$ to be loaded in shared memory.
20:         Load $L_i, D_i$ from HBM to on-chip SRAM.
21:         On chip, compute $\mathbf{S}_i^{(j)} = \mathbf{Q}_i \mathbf{K}_j^T \in \mathbb{R}^{B_r \times B_c}$ (SS-GEMM). Commit.
22:         Wait for $\mathbf{dO}_i$ to be loaded in shared memory.
23:         On chip, compute $\mathbf{dP}_i^{(j)} = \mathbf{dO}_i \mathbf{V}_j^\top \in \mathbb{R}^{B_r \times B_c}$ (SS-GEMM). Commit.
24:         On chip, wait for $\mathbf{S}_i^{(j)}$, then compute $\mathbf{P}_i^{(j)} = \exp(\mathbf{S}_{ij} - L_i) \in \mathbb{R}^{B_r \times B_c}$.
25:         On chip, wait for $\mathbf{dP}_i^{(j)}$, then compute $\mathbf{dS}_i^{(j)} = \mathbf{P}_i^{(j)} \circ (\mathbf{dP}_i^{(j)} - D_i) \in \mathbb{R}^{B_r \times B_c}$.
26:         On chip, compute $\mathbf{dV}_j \leftarrow \mathbf{dV}_j + (\mathbf{P}_i^{(j)})^\top \mathbf{dO}_i \in \mathbb{R}^{B_c \times d}$ (RS-GEMM). Commit.
27:         On chip, compute $\mathbf{dK}_j \leftarrow \mathbf{dK}_j + \mathbf{dS}_i^{(j)\top} \mathbf{Q}_i \in \mathbb{R}^{B_c \times d}$ (RS-GEMM). Commit and wait for both $\mathbf{dV}_j$ and $\mathbf{dK}_j$.
28:         On chip, compute $\mathbf{dQ}_i^{(\text{local})} = \mathbf{dS}_i^{(j)} \mathbf{K}_j \in \mathbb{R}^{B_r \times d}$ (SS-GEMM), and write $\mathbf{dQ}_i^{(\text{local})}$ to smem. Notify the $\mathbf{dQ}$-writer.
29:     **end for**
30: **else if** in $\mathbf{dQ}$-writer warp **then**
31:     **for** i in assigned Q-tile schedule **do** **[DASH]**
32:         Wait for $\mathbf{dQ}_i^{(\text{local})}$ to be ready in smem.
33:         Wait until the global order grants this block its turn to reduce. **[DASH]**
34:         Using a semaphore, atomically add $\mathbf{dQ}_i^{(\text{local})}$ to $\mathbf{dQ}_i$ in global memory.
35:         Advance the global order. **[DASH]**
36:     **end for**
37: **end if**

---

