# OpenReview forum: "DASH: Deterministic Attention Scheduling for High-throughput Reproducible LLM Training"
_ICLR.cc/2026/Conference — ICLR 2026 Poster_

### Official Review · Reviewer_aww5 · 2025-10-27

**Soundness:** 4
**Presentation:** 3
**Contribution:** 3
**Rating:** 4
**Confidence:** 4

**Summary:**

The paper focuses on improving the efficiency of the deterministic version of Flash-Attention (Backward). In FA, the backward involves an AtomicAdd operator to accumulate the dQ tensor. To make the kernel deterministic, it requires making the accumulation conducted in a fixed order. Existing methods introduce too many bubbles when handling this. The authors reframe this as a DAG scheduling problem and introduce DASH, a framework with two new scheduling strategies. Their methods improve throughput by up to 1.28x over the deterministic FlashAttention baseline by minimizing pipeline stalls and optimizing task execution order.

**Strengths:**

1. The presentation is clear and intuitive. 1.28x speedup is also quite practical.
2. The proposed method, Shift Scheduling, is clear to understand and can be easily proved to be optimal.

**Weaknesses:**

1. "Optimal" Method is Sub-Optimal for Most State-of-the-Art Models: The paper's core theoretical contribution, the "Symmetric Shift Scheduling," is shown to be less performant than the simple baseline when headdim>=128 (Figure 9b), a degradation attributed to register pressure. This is not a minor edge case: it is the standard configuration for a vast majority of today's most influential and widely used large language models, such as Llama 3, Qwen 3, Mixtral 8x7B, and DeepSeek, all of which use a head dimension >= 128. This severely limits the practical applicability of what is presented as a key contribution of the work.
2. Scalability to Long-Context Scenarios is Questionable: the "Shift Scheduling" method's performance advantage diminishes with sequence length, and it ultimately becomes slower than the baseline at the maximum tested length of 16,384 tokens.

**Questions:**

See weakness.

---

> ### Author Response · Authors · 2025-11-18
>
> Thank you for your constructive and thoughtful review. We appreciate your recognition of our contributions, as well as your valuable suggestions and concerns regarding the practicality and scalability of our scheduling strategies. We address your specific points in the following responses.
>
> ### 1. On the “optimal” schedule being sub-optimal at headdim ≥ 128
>
> We appreciate the reviewer highlighting this point. The observed performance drop in Figure 9(b) is attributable to register pressure in the underlying FA-3 fused kernel, rather than to any intrinsic limitation of the Symmetric Shift algorithm. FA-3 already utilizes nearly the maximum allowed registers per thread, owing to its highly optimized non-deterministic implementation. The Symmetric Shift schedule introduces approximately ten additional bookkeeping registers, which are sufficient to exceed the hardware limit on Hopper GPUs and cause register spilling to local memory. These spill-induced stalls dominate the runtime and obscure the expected benefits of balanced scheduling.
>
> Importantly, Symmetric Shift is not slower than the “simple baseline”; rather, its performance is only marginally behind that of Descending Q-Tile Iteration (DQTI), another scheduling strategy we propose that is specifically tailored to circumvent FA-3’s register limitations. For current GPU architectures, DQTI therefore represents the preferred scheduling approach for large head dimensions.
>
> Nevertheless, Symmetric Shift remains significant as the optimal critical-path schedule within our DAG model. Its theoretical advantages are expected to be fully realized on newer architectures with greater on-chip resources (such as Blackwell GPUs with TMEM, or devices equipped with larger register files), or under kernel designs that are less constrained by register allocation than the present FA-3 implementation. We have clarified the issue in Section 4.3 of the revised manuscript.
>
> ### 2. On the diminishing advantage at long sequence lengths
>
> We agree that the gains of Shift Scheduling decrease at very long sequence lengths. This behavior is expected and follows directly from the analysis in Section 3.2. Under a full attention mask, the deterministic baseline incurs only a fixed startup bubble $(n-1)r$, where $n$ is the number of KV tiles and $r$ is the reduction time, which does not grow with sequence length. As the number of tiles increases, this fixed cost is rapidly amortized, and the baseline naturally approaches the theoretical lower bound of our DAG model. At seqlen = 16,384, the baseline already achieves over 90% of the non-deterministic throughput, leaving very limited room for improvement from any schedule.
>
> In this near-saturated regime, small hardware effects, especially cross-SM synchronization via remote L2 segments, can outweigh the benefits of a more balanced execution order, causing Shift Scheduling to appear marginally slower. This is a hardware-driven effect under extreme parallelism rather than a scalability limitation of the scheduling algorithm.
>
> We also note that most full-mask workloads in practice (e.g., ViT, StableDiffusion-3.5, masked-LM models such as BERT and LLaDA) operate at moderate sequence lengths (≤4k), where the startup bubble remains significant and DASH consistently provides improvement. Only certain video-centric DiT models routinely reach >100k tokens, where the deterministic baseline already operates near the hardware limit.

---

> > ### Comment · Reviewer_aww5 · 2025-11-21
> >
> > Thank you for your explanation. My concerns are addressed, and I have raised my score.

---

> > > ### Author Response · Authors · 2025-11-22
> > >
> > > We thank Reviewer aww5 for the careful reconsideration of our clarifications and for the updated evaluation. We appreciate the reviewer's thoughtful engagement and the time and attention devoted to our submission.

---

### Official Review · Reviewer_rDjX · 2025-10-30

**Soundness:** 4
**Presentation:** 4
**Contribution:** 2
**Rating:** 6
**Confidence:** 4

**Summary:**

DASH is a kernel implementation for attention that allows deterministic backward passes with less overhead, compared to the default non-reproducible kernels. To achieve determinism reduction operations are ordered, which incurs overhead. To reduce them, DASH employs two techniques. First, it fills the pipeline bubbles caused by the irregular ordering pattern of causal attention with the next attention heads ordered in reverse. Second, it formalizes reduction operations through a DAG of operations, and reorders the deterministic order so the critical path is minimized. DASH is tested on an H800 GPU and compared to a deterministic version of Flash-Attention 3, and achieves higher throughput on a variety of sequence lengths.

**Strengths:**

Thank you for submitting your work.
- I'd like to praise the authors for their presentation. This was a pleasure to read. I was astonished how such a dense topic was explained so well in the text.
- FlashAttention 3 is a strong baseline, and the NVIDIA H800 GPU serves as a typical setup.
- The ideas are sound and the reason they occur is intuitive, though less so for shift scheduling.

**Weaknesses:**

- Though a minor point, I think there is merit to discussing the effect of determinism in other operations of the transformer. Matrix multiplications include reductions as well, correct? Does determinism reduce performance there as well?
- I think the paper should include a table that shows what the end-to-end relative benefit is for **a whole transformer block**, not just the attention part. While there is value in making attention faster, it is hard to put these gains in perspective without seeing the whole picture. For the transformer block configuration, pick a popular LLM, e.g., Llama3.

**Questions:**

Please see the weaknesses.

---

> ### Author Response · Authors · 2025-11-18
>
> Thank you very much for your thoughtful review and for your kind words about the clarity and presentation of our work. We appreciate your constructive feedback and address your concerns point-by-point below.
>
> ### 1. On determinism in other transformer operations
>
> Other components, such as GEMMs and normalization, also involve reductions, but in typical LLM training workflows, enforcing determinism incurs minimal overhead. GEMMs generally remain deterministic unless the reduction axis (K-dimension) is split across blocks (split-K/stream-K), which is rarely required given sufficient parallelism along M/N dimensions in large-batch scenarios. As a result, disabling split-K/stream-K has little impact on overall throughput.
>
> Reductions in LayerNorm, attention forward passes, and similar operations are performed within a single block and follow a fixed execution order, ensuring determinism. Additionally, purely elementwise operations, such as activations or bias additions, are inherently deterministic. We have clarified these points in Section 2.3.
>
> ### 2. On end-to-end impact at the transformer-block level
>
> We appreciate the suggestion to contextualize kernel-level improvements at the transformer block level. In response, we have added Figure 10 in the revised manuscript, which presents end-to-end speedup results for several representative transformer models, including LLaMA3-8b, Qwen2.5-7b, Mistral-8×7b, SAM-huge, StableDiffusion-3.5, and LLaDA. Across these models, we observe an average speedup of approximately 5%, consistent with our internal measurements during large-scale production training.

---

### Official Review · Reviewer_DP1T · 2025-11-01

**Soundness:** 3
**Presentation:** 3
**Contribution:** 3
**Rating:** 6
**Confidence:** 3

**Summary:**

This paper introduces DASH (Deterministic Attention Scheduling for High-Throughput), a framework addressing the performance loss in deterministic backward passes of attention mechanisms, particularly FlashAttention-3. Determinism ensures reproducibility in LLM training but enforces sequential gradient accumulation, limiting GPU utilization.

DASH formalizes the backward pass as a DAG scheduling problem, minimizing critical path length. Two strategies are proposed:

- Descending Q-Tile Iteration (DQTI): reverses query-tile traversal to reduce pipeline bubbles in causal attention.

- Shift Scheduling: a theoretically optimal schedule ensuring conflict-free accumulation, extended as Symmetric Shift Scheduling for causal masks.

Experiments on NVIDIA H800 GPUs show up to 1.28× speedup over deterministic FlashAttention-3. However, for long sequences or large head dimensions, inter-SM communication latency and register pressure diminish the theoretical gains.

**Strengths:**

- Novel kernel implementation for an important operation in deterministic Transformer training
- Clear and novel DAG-based formalization of deterministic backward scheduling.
- The two scheduling strategies are well-motivated, combining theory and practicality.
- Empirical validation on modern GPUs with thorough analysis of full vs. causal masks.
- Addresses an important reproducibility issue in large-scale deterministic training.

**Weaknesses:**

- There is performance degradation for long sequence length with full attention mask (Figure 8)

- The theoretical model ignores some of the GPU implementation considerations, such as inter-SM communication overhead and register requirements (as mentioned in sections 4.2 and 4.3)

- Focuses solely on the backward pass; potential extensions to the forward path are not explored.

- Symmetric Shift Scheduling introduces significant register pressure, limiting practical benefits.

**Questions:**

- Do we really need sequential accumulation to ensure reproducibility? Can we do some sort of parallel reduction like in the binary tree to achieve better efficiency with determinism?

- Are there similar challenges for the forward operation of attention?

---

> ### Author Response · Authors · 2025-11-18
>
> We thank Reviewer DP1T for their thoughtful review and valuable feedback on our work. We appreciate the recognition of our contributions and now address the reviewer’s questions and concerns below.
>
> ### 1. Performance degradation at long sequence length (full mask)
>
> This observed behavior aligns with our analysis in Section 3.2. Under a full attention mask, the baseline schedule already establishes a steady pipeline. The main additional cost from enforcing determinism is the fixed startup bubble, $(n−1)\cdot r$, which does not increase with sequence length and is quickly amortized as the number of tiles grows. As sequence length increases, the baseline’s throughput approaches the theoretical lower bound predicted by our DAG model. For long contexts (e.g., seqlen = 16,384), the deterministic backward pass sustains over 90% of the non-deterministic throughput, meaning any further scheduling improvements can only recover a marginal gap.
>
> As the baseline is near-optimal, further improvements are fundamentally constrained, and practical hardware factors such as inter-SM communication latency (Section 4.2) may even cause minor overheads.
>
> Notably, most full-mask applications in practice (e.g., ViT, StableDiffusion-3.5, masked language models like BERT and LLaDA) operate at moderate sequence lengths ($\leq$ 4096), so DASH consistently improves performance in these scenarios. Only certain video-centric models (e.g., DiT architectures processing >100K tokens) regularly reach the regime where deterministic baselines already approach hardware limits.
>
> ### 2. Theoretical model vs hardware considerations
>
> Our theoretical model deliberately abstracts away hardware-specific details such as inter-SM communication overhead and register requirements, with the aim of deriving a closed-form, portable scheduling policy. Integrating inter-SM latency into the model would entangle the schedule with hardware parameters and input sizes, making the problem highly hardware-dependent and unlikely to yield closed-form solutions. Addressing detailed register usage would also require low-level simulation, which often adds complexity without substantially benefiting algorithm design. By focusing on an abstract model, we aim to present scheduling principles that are widely applicable, rather than limiting insights to a specific GPU architecture. We further clarified the design rationale behind our model in Section 3.1 in the revision.
>
> ### 3. Is sequential accumulation necessary for determinism?
>
> In principle, determinism does not require strictly sequential accumulation: a binary-tree reduction is also deterministic if its order is fixed. The difficulty in FlashAttention-style backward kernels is locality: dQ only resides transiently on-chip, so each tile’s partial dQ must be finalized before we advance to the next tile. Building a deterministic tree across tiles, therefore, needs a separate post-reduction kernel that first writes all tile-level partials to global memory and then reads them back to produce the final dQ, as in the deterministic mode of FA-2. This introduces an extra full pass over dQ and an additional kernel launch. By contrast, DASH only changes the in-kernel update order so that deterministic accumulation completes in place within the original backward kernel.
>
> ### 4. Are there similar issues in the forward pass?
>
> Similar issues do not occur in the forward pass. In this case, each CTA processes a Q-tile independently, and reductions over KV tiles are performed entirely within a single CTA in a fixed order, eliminating the need for inter-SM accumulation. Consequently, the forward kernel is inherently deterministic. By contrast, nondeterminism arises exclusively during the backward pass due to cross-SM reductions required for computing dQ. We added a clarifying discussion in Section 2.3 in the revision.
>
> ### 5. Register pressure in Symmetric Shift Scheduling
>
> The observed performance degradation primarily results from the underlying FA-3 kernel implementation rather than the scheduling algorithm itself. FA-3 operates very near the register limit due to aggressive kernel fusion. While our scheduling approach introduces only about 10 additional bookkeeping registers, a small amount compared to the number of registers available per thread on Hopper GPUs, this modest increase is still sufficient to trigger register spilling on Hopper. Notably, the scheduling algorithm itself remains lightweight.
>
> Despite these implementation-specific limitations, Symmetric Shift Scheduling retains practical value as the optimal critical-path solution. On hardware with larger on-chip resources (such as Blackwell GPUs with TMEM or architectures offering expanded register files), or with less register-constrained kernel designs, its theoretical benefits can be fully realized. For contemporary GPU architectures, we therefore recommend Descending Q-Tile Iteration as a more practical solution for large head dimensions.

---

### Official Review · Reviewer_RNMq · 2025-11-04

**Soundness:** 3
**Presentation:** 4
**Contribution:** 3
**Rating:** 8
**Confidence:** 3

**Summary:**

1. The paper builds on identifying the cause of nondeterminism in Flash Attention to be the atomicAdd for the dQ reduction, and identifies pipeline bubbles caused by the default ordering of processing the reduction, showing that it is specially prevalant in the case of causal attention because of the dependency order enforced by the causal mask.

2. The authors propose a simple heuristic of reversing the order of the query blocks on the SMs and interleaving the processing of different heads to reduce the pipeline bubbles.

3. The authors then model the scheduling problem as a critical path identification problem on a DAG, with within SM computations and reductions forming an isomorphic chain, and data dependencies across SMs (dQ transfer) as zero weight dependencies, and using that to inform the schedule to minimize critical path, thereby demonstrating strong gains over baseline deterministic variant of FA3.

**Strengths:**

1. The paper addresses an important area of deterministic training that has been gaining a lot of traction especially with large model training, bridging the gap between the non deterministic and the deterministic version of the attention kernel, which serves as one of the fundamental pieces in the commonly used transformer architectures.

2. I especially like the in-depth analysis of when the theoretical model deviates from the on hardware execution results: both for scenarios with long context length for no mask scenario where the inter-SM communication latency starts becoming the bottleneck, and for the large head size analysis with the causal mask.

**Weaknesses:**

My only concern is that given the motivating backward's schedule analysis presented in Section 3.2, I would have expected the deterministic attention baseline to have been more competitive with shift scheduling for the non causal mask scenario.

Similarly, I would have expected the descending schedule to have been much better compared to baseline for the causal mask case with head size 64. That does not seem to be the case. Would it be possible for the authors to specify what might be the cause of it, since without that, it's a bit difficult to verify the correctness of the theoretical model, which is the underpinning of the proposed approach.

**Questions:**

Besides my primary concern above, I had two other questions:

1. The main issue for why we are unable to capture the long sequence length issue in the scheduling for the non causal mask setup seems to be the assumption of the zero weighted edge. Is that not something that can be factored in (eg: a cost reduction optimization on the DAG with weighted edges ?)

2. [Minor] Given the popularity of FP8, would it be possible to also compare against FA3 on the a similar setup, just get get more confidance on the algorithm's applicability for lower precision training ?

Minor Presentation Nits:

The authors go from representing a compute reduction blocks in the Gantt chart as a function of (kV index, query index) in Figure 2 to a (query index) representation in Figure 3. It is a bit confusing, so would be good to be consistent.

It would be good to include the exact algorithm (similar to how it's presented in FA3) for the proposed backward pass

---

> ### Author Response · Authors · 2025-11-18
>
> We thank Reviewer RNMq for the detailed and insightful review, as well as the balanced assessment of our work’s strengths and areas for improvement. We address each of your concerns in detail below.
>
> ### 1. Why the deterministic baseline remains competitive under the non-causal mask
>
> Our DAG analysis in Section 3.2 gives, for the full mask, $T_{\text{full}} = m\,n(c+r) + (n-1)r,\quad T_{\text{full}}^{\text{opt}} = m\,n(c+r)$, where n is the number of KV-tiles, m is the number of heads, r and c are the respective times for reduce and compute. Thus, the only gap to the optimal schedule is the startup bubble $(n-1)r$. In one configuration in Fig. 8 (headdim = 64, seqlen = 8192, KV-tile size = 128, hence n=64), we measure a compute–reduction ratio of about $c:r \approx 5:3$. Plugging these numbers into the model gives a per-head upper bound of roughly $1.37\times$. The expression with the factor m in section 3.2 assumes that all m heads share one global startup phase, so the bubble is amortized over all heads, and the maximum kernel-level speedup becomes $1 + \frac{(n-1)r}{m\,n(c+r)}$, which is ≈1.006× when m=64.
>
> In practice, however, heads are executed in parallel across 132 SMs, and each head has only 64 KV tiles, so 64 does not divide 132. This leads to partially filled waves of CTAs: the pipeline “ramps up” multiple times across different head groups, and the startup bubble is only amortized over a small effective number of heads per wave rather than all 64. Consequently, the effective startup overhead is larger than what the idealized fully serialized model (with a single global ramp-up) would predict, leaving only a modest but non-negligible headroom that Shift Scheduling can exploit.
>
> Moreover, in the baseline implementation, many CTAs busy-wait on the same global counter via acquire loads, and the serialized atomicAdds on that counter create a hotspot in the distributed L2 and amplify inter-SM synchronization costs, especially when the compute phase is short.
>
> Together, these effects explain why the deterministic baseline remains competitive under the full mask yet still leaves a modest performance margin for Shift Scheduling to exploit.
>
> ### 2. Why Descending yields limited gain at headdim = 64 for causal attention
>
> In the FA-3 causal backward kernel, the L2-aware LPT scheduler interleaves multiple heads across SMs. When headdim = 64 and the sequence length is short, each head’s L2 footprint remains small, allowing many heads to reside in cache with only 1–2 tiles in flight per head. Consequently, the causal stalls targeted by Descending Q-Tile Iteration are largely masked by cross-head interleaving, resulting in only marginal net performance gains. This is evident from the baseline TFLOPS results: for causal attention at seqlen = 2048, headdim = 64 achieves 334.4 TFLOPS, slightly outperforming the 311.4 TFLOPS at headdim = 128, despite higher per-head compute at the latter. As the headdim increases to 128, the per-head L2 footprint approximately doubles, reducing the number of heads that can be effectively interleaved. Execution becomes increasingly dominated by per-head pipeline lag, and Descending Q-Tile Iteration yields significantly greater improvements under these conditions. We clarified this behavior in section 4.3 in the revision.
>
> ### 3. On the use of zero-weight edges in the DAG model
>
> We represent inter-tile dependencies as zero-weight edges to facilitate the derivation of a closed-form, implementable scheduling strategy, rather than to exactly mirror hardware-specific runtimes. Adding weighted edges would tightly couple the scheduling order with input dimensions and inter-SM/L2 cache latencies, making the problem excessively complex and hardware-dependent, and precluding closed-form analysis. Such complexity would obscure essential scheduling insights and hinder portability across GPU architectures. We therefore adopt a deliberately simplified model for analytical clarity and evaluate its limitations, such as long-sequence deviations from L2 segmentation, empirically in Section 4.2. We will make this modeling choice and its rationale more explicit in Section 3.1.
>
> ### 4. On FP8 comparison
>
> FA-3 does not include an FP8 backward kernel, and our current implementation is based directly on FA-3. Consequently, we are unable to provide deterministic FP8 backward results at this time. Nonetheless, our proposed scheduling algorithms are arithmetic format-agnostic and will be directly applicable once an FP8 backward pass is implemented.
>
> ### 5. Presentation suggestions
>
> We appreciate the reviewer’s suggestions and have updated Figure 3’s caption to explicitly explain the omission of KV indices, noting that this decision was made due to space constraints. In addition, we have incorporated the full backward algorithm in Appendix C, as requested, to further enhance clarity.

---

### Meta-Review · Area_Chair_xSMM · 2026-01-05

**Summary:**

The manuscript addresses the inefficiency issue in the current attention implementations, due to the suboptimal scheduling of compute and gradient‑reduction phases. It formulates the backward pass of deterministic attention as a scheduling problem on a Directed Acyclic Graph (DAG) and derives schedules that minimize the critical path length.

All reviewers were satisfied with the manuscript and reached a consensus of acceptance.

**Reviewer Concerns:**

Reviewer RNMq expects that (1) the deterministic attention baseline will be more competitive with shift scheduling for the non-causal mask scenario; (2) the descending schedule should be much better compared to baseline for the causal mask case with head size 64.

Reviewer DP1T concerns (1) the performance degradation for long sequence length with a full attention mask; (2) the theoretical model ignores some of the GPU implementation considerations; (3) focuses solely on the backward pass; (4) symmetric shift scheduling introduces significant register pressure.

Reviewer rDjX encourages discussing (1) the effect of determinism in other operations of the transformer; (2) including a table to show what the end-to-end relative benefit is for a whole transformer block, not just the attention part.

Reviewer aww5 concerns that (1) the "Optimal" Method is Sub-Optimal for Most State-of-the-Art Models; (2) Scalability to Long-Context Scenarios.

**Reviewer Scores:**

Reviewer aww5 had raised the score from 4 to 6.

---

### Decision · Program_Chairs · 2026-01-26

Accept (Poster)